# Healthy Long-Lived Human Beings—Working on Life Stages to Break the Limitation of Human Lifespans

**DOI:** 10.3390/biology11050656

**Published:** 2022-04-24

**Authors:** Weikuan Gu

**Affiliations:** 1Department of Orthopedic Surgery and BME-Campbell Clinic, University of Tennessee Health Science Center, Memphis, TN 38163, USA; wgu@uthsc.edu; Tel.: +1-901-448-2259; 2Research Service, Memphis VA Medical Center, 1030 Jefferson Avenue, Memphis, TN 38104, USA; 3Department of Pharmaceutical Sciences, University of Tennessee Health Science Center, 881 Madison Ave, Memphis, TN 38163, USA

**Keywords:** aging, castration, life cycle, lifespan, life stage, longevity, oophorectomy, puberty, reproductivity

## Abstract

**Simple Summary:**

This paper proposes a hypothesis that the human lifespan could be extremely long and healthy if different life stages were elongated with specific strategies. If the hypothesis is tested and confirmed, biomedical research will enter a new era and the dream of long-lived human beings may be achieved in the near future.

**Abstract:**

The human lifespan has been increasing but will soon reach a plateau. A new direction based on the principal law of lifespan (PLOSP) may enable the human lifespan to be extremely healthy and long by the proper manipulation of the well-defined growing stages of the lifespan. The lifespan of creatures on earth from a single cell to animals can be elongated at different life stages including prenatal development, body growth, reproductivity, and aging. Each life stage has its own specific physiological and metabolic characteristics. Each life stage can be lengthened by either slowing its processes or continuously maintaining the activities of its function. Unfortunately, the current biomedical research on the extension of lifespans has mainly focused on the aging stage. Recognizing and clearly defining the periods of transition and the boundaries of life stages are essential for achieving the goal of long-lived healthy humans based on the PLOSP. The biomedical measures and pharmacological treatments for the extension of lifespans is life-stage-specific. The PLOSP can be tested with modified studies on longevity with a variety of technologies such as castration and ovariectomy. Sex differences in biological functions and the sequential order of the life stages requires different approaches for females and males.

## 1. Introduction—Breaking the Limitation on the Longevity of Human Beings

How soon humans can live up to or surpass the highest limit of their lifespans is the question to which everyone wants the answer. Although human longevity has been increasing, there are several unpleasant aspects that discourage the public as well as researchers. First of all, while the longevity of the human population has increased around the world, the speed of this increase has been slow. For example, the life expectancy of the average American increased from about 45 years in 1900 to over 77 years at present [1,2]. Thus, it took more than a century to increase the lifespan by 32 years. In fact, the annual growth rate has decreased from approximately 0.3% 50 years ago to approximately 0.05% in recently years. Based on this trend, it may take more than 200 years before an American can live up to 100 years. Secondly, there have been controversial effects reported for castration and ovariectomy in animal models, causing confusion as well as a loss of confidence on these approaches [3,4,5,6,7]. Furthermore, it has been reported that human longevity has an upper limit. According to the report, under completely ideal biological conditions, blood markers of aging would simultaneously diverge at a critical point of 120–150 years of age, so that they could no longer support a living organism [8]. Thus, regardless of the speed of progress on longevity, the human lifespan will stop at 150 years. 

The discouragement is partially caused by ignoring the fundamental principle of the lifespans of living organisms. Theoretically, humans can live for an extremely long or unlimited length of time by the proper manipulation of the growing stages of the life cycle. It is time to carefully examine the principal law of lifespan (PLOSP) for long-lived healthy human beings. Because this is the first time the concept of the PLOSP has been introduced, a detailed explanation of the PLOSP and a discussion on life cycles and lifespans will be provided in this article. The purpose of this article is to bring attention to how, based on the PLOSP, extremely long and healthy lifespans for humans could possibly be achieved. 

## 2. Evidence of Long Lifespans with Variations to Stages of Life Cycles of Different Organisms That Support the PLOSP

Every living creature has its own life cycle. Changes to the length of the stages in this life cycle may change the lifespan. Due to the nature of genetic material, every living creature has its own features within its life cycle, including birth, growth, reproduction, and death [9]. The total length of all these stages determines the lifespan. Each life stage has its own characterization in terms of physiological and metabolic activities. The following are easy-to-understand and common-knowledge illustrations of long lifespans achieved by variations to the stages of the life cycles of different organisms. 

### 2.1. The Basic Life Cycle and Lifespan Foundation in Single Cell Organisms

Understanding the PLOSP can be started by reviewing the essential component of life, the single cell [10]. The stages of a life cycle of a cell in a growing tissue include growth, DNA synthesis, predivision and continued growth, and mitosis (the end of the life of the original cell) (Figure 1A). At the cell growth stage, cells increase in size and DNA copies, store energy, biosynthesize materials, and synthesize growth factors. At the division stage, the cell vanishes, while materials are divided into two new cells.

Similar pattens occur in simple lives on earth. Bacteria progress through four stages of growth: the lag phase, the log phase, the stationary phase, and the death phase (Figure 1A) [11]. Most indoor fungi (molds) go through a four-stage life cycle: spore, germ, hypha, and mature mycelium [12]. These simple living creatures depend on reproduction for their survival as colony populations. For each individual simple cell, the stopping of any stage of its life cycle means death. Extending any stage of the life cycle means an increase in the lifespan. 

If we consider mitosis and division as the end of the life of a cell or bacteria, increasing the lifespan depends on the growth and synthesis stages (Figure 1B). 

### 2.2. The Limitations and Breaking the Limitations to the Life Stages in the Lifespan of Plants

The flowering of certain plants heralds death. Otherwise, the plant continues to grow [13]. Modern agriculture grows varieties of crops that mature at the designated times. During vegetative growth, a plant focuses on body enlargement, such as leaf, stem, and root growth, with an elaborate program of cortical morphogenesis that replicates the cortical organelles. The turning point of the lifespan of a crop is the change from the vegetative to the reproductive stage. When such a transition happens at the right time, the crop becomes mature in the expected timeframe. The maturity of a crop means the stopping of its physiological and metabolic activity. However, this is not true for every individual crop. The environment can cause the elongation of vegetive growth, so the crop will delay its time of maturity (Figure 2). Many crops, such as wheat and cotton, will grow wildly when excessive nitrogen fertilizer is used, delaying fruiting and maturity and resulting in prolonged growth periods (Figure 2A). In other words, prolonging the growth of the plant body also prolongs the lifespan of the individual plant.

Among vegetables, a typical example that I have come across is the pepper. The little chilies are green at first. When all the small peppers over the whole plant body turn red, they have reached the stage of maturity. After that, the individual small pepper plant withers and dies. However, in some cases, the little peppers are kept growing green all the time. This is achieved by removing all the little peppers before they turn red. As long as the small peppers are removed before they turn red, the plant will continue to grow and continue to produce small green peppers (Figure 2B). This example shows that extending the production period of its fruit-bearing can also extend the life of the individual plant.

Thus, for plants, the lifespan is determined by certain circumstances. 

Increased lifespan = storing the seeds for a longer time while keeping them alive with more suitable conditions + the extension of vegetative growth + slowing down the flowering or keeping the plant at the reproductive stage (Figure 2B). 

Shortened lifespan = immediate seeding + fast physiological growth + rapid maturation (Figure 2A).

Killing a plant = consuming the seeds or destroying the growing plant at any time (Figure 2A).

### 2.3. The Life Cycle of Insects and Variations to the Stages of Their Lifespans

The life of a butterfly or moth includes four stages: fertilized egg → larva → pupa → adult (Figure 3) [14]. On the other hand, a grasshopper lives in three stages: egg, nymph, and adult. One important fact is that after an adult (butterfly or moth) breaks out of its cocoon, it lays eggs and then dies immediately or in a couple of days. Thus, for these insects, the accomplishment of reproduction means the exhaustion of the body energy and death. In particular, the male adult will die after mating. Some insects only live for hours, while others live for years. Those who have long lifespans usually have a very long development period as immature larvae or nymphs (Figure 3A). This phenomenon is similar to the increase in the vegetative stage in plants. The lifespan of insects can vary greatly depending on the environmental conditions such as temperature, humidity, day length, and the amount or quality of food available. Similarly, for insects, the lifespan depends on the length of the different stages in its life cycle (Figure 3B). 

### 2.4. The Variation of Lifespan Caused by the Longer Duration of Body Development and Reproductivities Stages in Animals

Among animals, those with a relatively long lifespan are basically divided into two categories. In one category are those with a relatively long duration for all stages of growth and development. That is, the physiological and metabolic activities of body growth are slow; e.g., Greenland shark and tortoises [15]. In the other category are those who maintain productivity or maintain prolonged physiological and metabolic activities at high levels for a long time during the reproductive stage, such as ocean quahogs [16] and koi fish. The aging or dying stages do not constitute a major portion of the lifespan of animals. 

The longevity of body growth can be achieved in two ways: continuous growth into a large body size or a slow rate of growth. For example, the Greenland shark has the longest known lifespan of all vertebrates [15]. Females may not reach sexual maturity until they are 100 to 150 years old (Figure 4A). Tuataras with a small body size mature slowly and do not stop growing until they reach about 30 years old. Their lifespan is around 60 years and it is thought that they can live for up to 100 years in the wild [17]. 

One example of an animal with a long reproductive period is the ocean quahog. It has been reported that the male reaches sexual maturity age at 10 years old and the female at 13 years old [16] (Figure 4A). However, many of them will live to see their 400th birthday, and the oldest on record was 507 years old when it was caught off the coast of Iceland in 2006 [18]. Another example is the Japanese koi, which on average lives for around 40 years, though they can live a lot longer. Koi fish are considered sexually mature when they are 2 years old, but koi are able to produce baby koi fish until they are up to 15 years old [19]. Thus, their longevity is dependent on their long period of reproductivities.

The Labord’s chameleon has the shortest lifespan of all land vertebrates (Figure 4B)—it dies in a year [20]. The lifespan of the small mosquitofish only reaches 1.5 years. The common characteristics between these two animals are that they grow extremely rapidly in all life stages, reach sexual maturity, breed, and then die. In addition, some organisms can only produce offspring once during their reproductive stage, such as Chinook salmon [21], which reproduce and then die.

### 2.5. Sex Difference and the Short Lifespan of Males

The aforementioned data on the lifespans of plants, insects, and animals are mostly applicable to females. The style and variations of the lifespans of males are in many cases different from those of females. In general, the lifespans of males are shorter than those of females. In particular, there are many literature resources reporting the death of males after mating [22]. For example, the male honeybee dies soon after mating. Its lifespan is much shorter than that of the queen. The male dark fishing spider dies instantly following mating. The male little red kaluta’s immune systems collapses, and it dies of stress-related issues after mating. The male antechinus dies after nonstop 14-h sex sessions. In most of these cases, the death of the male is the result of the exhaustion of the body’s energy and the deterioration of the organs. 

### 2.6. Different Organisms Achieve a Long Life with Different Elongated Stages

In summary, different organisms achieve a long life with different elongated stages in their life cycles (Figure 5A). The body sizes of long-lived organisms are either larger or smaller than those of humans. Thus, regardless of the variations in body size, these organisms have a lifespan longer than humans. Therefore, there is no reasonable argument that humans could not live longer than these organisms on the earth.

## 3. The Current Research Has Not Followed the PLOSP for Organisms

### 3.1. The Aging Stage Has Been the Major Focus in the Study of Longevity for Humans

Given the fact that we human beings have developed a relatively high level of intelligence and understand significantly the genomic components of ourselves and other organisms, why has our lifespan not been extended much? Ironically, the most likely answer is that our research on longevity has not been in the right direction, and the differences in between the various life stages have not been well incorporated in our research. For example, studies on healthy nutrition have never clearly been divided according to life stages such as the prepuberty, reproductive, and post-menopause stages. Studies on longevity have focused on the aging stage, which is the smallest portion of the lifespan of living creatures [23]. There is nothing wrong with saving people’s lives with biomedical technology. However, extending the human lifespan by saving aged persons’ lives is different from the elongation of human life by the extension of other life stages. Saving an aged person’s life is similar to saving an energy-exhausted crop or animal. The intentional extension of different life stages has never been the clear objective of the study of human longevity (Figure 5B). Typical examples of studies without the clear objective of lifespan extension are the investigations into oophorectomy and castration. 

### 3.2. Lessons from Oophorectomy in Females 

The study of oophorectomy started with the treatment of breast cancer. One of the most direct side effects of ovariectomy in patients with breast tumors or ovarian cancer is the aging of the body and the development of many diseases [24,25,26]. Hundreds of studies have been conducted on the diseases and molecular pathways related to oophorectomy, though rarely does anyone consider the human life cycle and that the removal of the ovary probably disrupts the normal physiological and metabolic processes in the reproductive stage of life and pushes the human body into an early post-reproductive or aging stage (Figure 5C).

Ironically, without realizing that the life cycle is in play, studies from animal models contradict those in humans, causing confusion in the study field. For example, while studies have indicated that the lifespan of ovariectomized females is short [27], one recent study reported that ovariectomized females have improved survival [28]. Looking at the study procedure in detail, it was indicated that the ovariectomy carried out at 5 months of age in mice shortened their lifespan, while the ovariectomies for mice with extended lifespans were carried out at 3–4 weeks of age. Thus, the extended lifespan was because the ovariectomy was performed at the time of prepuberty. An ovariectomy before prepuberty may slow down the sexual physiological metabolism and body growth, therefore extending the lifespan. 

If the lifespan theory is used to analyze the data from these studies, the conclusion is that the removal of the ovaries at an early enough time can slow down the transition from the pre-sexually mature stage to the sexually mature stage, thus delaying the maturity of the mice so that the lifespan is extended. On the contrary, after sexual maturity, maintaining the activity of the ovaries can delay the aging of mice. The lifespan theory is supported by a recent study of Mason et al. [29], in which the authors increased the lifespan of old mice by transplanting into them young ovaries. Mice that received germ-cell-depleted ovaries had extended lifespans [30].

There are a large number of studies on ovariectomies in animals [31]. However, the data are not useful because the time of the ovariectomy in animals is not clearly reported, only a small number of animals is included, or there is influence from other factors such a nutrition or drug treatment. 

### 3.3. Lessons from Castration and Eunuchs in Males 

There has been strong isolated evidence supporting the PLOSP. Unfortunately, this isolated evidence is not linked to the PLOSP. As early as in 1961, Robertson [29] reported the prolongation of the lifespan of kokanee salmon by castration before the beginning of gonad development. In 2012, the study of castration was encouraged again by the report that the average life expectancy of eunuchs is 14 to 19 years higher than that of ordinary people [3]. These important studies did not consolidate the research filed of longevity because of the lack of the concept of the PLOSP.

Without the concept of the PLOSP, castration was regarded as a technique or methodology mainly for the treatment of prostate cancer. Consequently, without realizing the critical issue of life stages, the side effects from castrated cancer patients were considered as evidence against a prolonged lifespan achieved by castration [4]. Similarly, in animal studies, castration did not show any benefit when the castration was performed around or after the time of puberty [5].

In contrast, in studies using animal models, when the castration was carried out during or before puberty, the lifespan was increased [6,32]. One clearer piece of evidence is that Sugrue et al. [7] reported that the castration of sheep at 5–50 days from birth, far before puberty, delays epigenetic aging. 

For dogs, the data is confusing, because the traditional age for neutering is 1 to 6 years of age [33], while male dogs can become sexually mature from 5 to 8 months of age. 

Similarly to ovariectomy, lifespans will most likely be prolonged if castration is performed before puberty but shortened if it is performed after puberty. Furthermore, the maintenance of the sex capability of the animal increases the lifespan. This is possibly true for eunuchs, cancer patients, and for animal models. 

### 3.4. Unrevealed Issues in the Transition Period from One Stage to the Next in the Life Cycle

The critical issue regarding to the transitions in the human life cycle from one stage to the next should have been recognized (Figure 5B). The first issue is related to when the right time is to slow down or stop sexual maturation, i.e., the time to perform castration and ovariectomy. The biological maturation inside the body most likely happens before the phenotypic appearance of puberty (Figure 5C). Although studies using mouse models conducted castration and ovariectomy a short time before or during the time of puberty, the best time to conduct these surgeries has never been investigated [3,7]. Similarly, in the data from humans, the time of castration for eunuchs ranged from early childhood to adult [3]. The neglect of such a key issue has led people to present their data without distinguishing the data from individuals before, during, and after puberty. It is very likely that the lack of such knowledge is one of the reasons that has led to the controversial results in the previous studies.

The second question is when the physiological transition from the reproductive stage to the post-reproductive or aging stage occurs. It is expected that the physiological activity inside the body occurs at an earlier time than the actual loss of reproductive ability. 

## 4. The PLOSP Can Be Demonstrated by Realizing the Extreme Length of the Proper Stages of the Life Cycle

Based on the evidence above and the PLOSP, the human lifespan can be extended to hundreds or thousands of years by a variety of measures for the elongation of one or more of the stages of the life cycle (Figure 5C). 

### 4.1. Extension of Gestation Stage May Be Useful

First of all, the question is whether it is necessary to extend the period of gestation, which is the time between conception and birth, that is, from the formation of the embryo to the birth. Although it may be practically difficult, and it seems useless to extend the human lifespan by increasing the length of time for which a baby is in their mother’s body, the extension of the length of this stage may affect the next stage, the body growth. The speed of body growth may be at least partially determined during gestation [34]. It is possible that slowing this process may lead to slow growth throughout the whole life (Figure 6A). 

### 4.2. Stage from Birth to Maturity of Reproductive Activity 

Perhaps the best way to extend this stage is to make our own bodies grow slowly. This period is for the accumulation and increasing of metabolic materials, the adjustment of enzymatic and hormonal activities, and the enlargement of organ and body size [35,36]. Of course, how many people are willing to live a long childhood remains a question. If we grow at a normal speed but only postpone sexual maturity, our body will grow too tall, from the current human body height of less than 2 m to more than 2 m, or even 3 m in height (Figure 6B–D). On the other hand, the extension of the lifespan of eunuchs and castrated mice suggest that men will be able to extend their lives for a certain period through delaying the time of puberty.

### 4.3. Stage from Sexual Maturity to Termination of Reproductivity 

New technologies will enable the elongation of this period in a variety of ways. The goal is to elongate the period of essential body functions, reproductivity, immune levels, normal circulation, and digestion [37]. A few examples will be given to achieve this goal. Furthermore, the methodologies for this stage between men and women may be different. 

Every woman is born with more than 30,000 basal egg cells. Among them, only 400–500 egg cells become mature and are discharged during ovulation [38]. Assuming that, under certain circumstances, the number of women’s ovulation cycles doubles, women’s average life expectancy could increase by about 30 years. If the number of women’s ovulation cycles continued to increase, then women could possibly become the evergreen little peppers (Figure 6B–D). 

In the case that it is impossible to increase the number of female egg cells, and the number of mature eggs in a woman’s life is fixed, we may try to slow down the ovulation period. In theory, there are different ways to achieve this. The first method is to extend the menstrual period from the current average of 27 days to two months or half a year. The second method is pregnancy. From conception to the birth of a newborn takes at least 10 months. If a female person becomes pregnant 10 times, then there will be 100 egg cells whose discharge is are postponed. Whether this will prolong the life of women can be answered by investigating the relationship between the number of children in a family and the mother’ lifespan. In addition, there is an issue as to whether a false ovulation period can be created as a condition that women can maintain through the duration of the reproductive period. With the development of medical science, there may be many ways, such as the use of genetic or physiological and biochemical methods, to extend the first life stage of women.

It may be more difficult to maintain men’s reproductive capability than women’s ovulation levels. From the plant to the animal world, in many cases, after males have completed their life-prolonging tasks, there is no need for survival. Compared with many other animals and plants, human males are already lucky. However, with the development of medical science, technology for the continuation of male reproductivity may appear. The positive ratio between the time of the male function of reproductivity and the lifespan may also be checked relatively easily through survey data from animal studies.

### 4.4. Stage of Aging—From End of Reproductivity to the End of Life 

Compared to the previous two stages, this stage is less favorable to be used for the extension of the lifespan, because this stage is the time of the deterioration of the body’s functions. However, most previous studies have focused on this period of time (Figure 6B–D). The changes in women at this stage are more obvious. How to keep the female body in the premenopausal physiological condition after menopause is a difficult research topic. Evidence from plants, insects, and animals suggest that it is not easy for men to maintain the same mental and physical conditions after they have completely lost their reproductive function. Thus, the strategy for the extension of lifespan for males relies heavily on the early growth and reproductive period.

### 4.5. New Technologies to Assist the Elongation of Lifespan

Tissue or organ regenerations may benefit from tree growth and other perennials or foliage plants [39,40]. Tissues stored at a younger stage of the life cycle may be used to replace or regenerate parts of a body at an older stage to extend its lifespan. This aspect is heavily dependent on the progress of medical sciences.

## 5. Challenges and Testing of the PLOSP

### 5.1. Challenges for the Extension of Lifespan 

The extension of lifespan based on the PLOSP may face challenges. At present, it is known that there are two challenges to the PLOSP. The first is that there are already reports on the potential upper limit of the human lifespan [8]. Regardless of the reliability and credibility of this research, we should at least realize that our current average human lifespan is only half of 150 years. Only if we double our lifespan can we reach 150 years. We will know whether such a limitation exists when we reach the lifespan of 150 years. It is believed that with the development of medical technology and science, the limit of the viability of blood cells could be surpassed or updated at any time. The second challenge is that with the extension of the lifespan, the probability of disease will increase. However, we should also realize that the incidence of disease is relatively low during the prosperous life stage. Many diseases occur due to the decline in human physiology, which stops the maintenance of vigorous human functions and the maintenance of immunity at a high level. The development of medical science and technology will support the postponement of life stages as well as diseases. 

### 5.2. Considerations for the Testing of the PLOSP

To test the PLOSP, one needs to understand the concept of life stages clearly. It is important to realize the difference between the physiological life stage and the phenotypic life stage. Thus, the sexually mature stage may happen long before the puberty phenomena. Therefore, researchers need to ensure that they group their study subjects correctly. The other consideration is that, currently, no specific medications or technologies can be safely used to extend a particular life stage. It may not be possible to conduct a study on the direct extension of a life stage. Therefore, directly testing the PLOSP by the extension of a life stage may not be possible. 

### 5.3. Proposed Potential Studies to Indirectly Test PLOSP

In the short term, several studies may be conducted to obtain data to test the PLOSP. First, we may revisit the data in the reported Korean eunuchs. Although the eunuchs were usually castrated at young age, the ages of castration varied greatly [3]. In the reported eunuchs, the lifespan also varied greatly; three of them lived longer than 100. If the PLOSP is correct, those castrated at time much earlier than the puberty age would have lived longer than those who were castrated late. 

Secondly, the data on neutered dogs and other animals can be reviewed. Some believe that neutered dogs and other animals that have had their sources of testosterone removed often live longer than their intact counterparts, while others do not. As the age of the neutering of the dogs varies greatly, they can be grouped into three groups: before puberty, after puberty, and unsure or during puberty. Those in the longer-life group are believed to be those neutered before puberty. 

A third simple study is the castration of laboratory mice. The mice can be divided into two groups: those castrated at a time much earlier than puberty, and those castrated after the puberty. The group with early castration should live longer than the later group.

## 6. Conclusions

It is expected that by the extension of a certain stage or stages of the life cycle based on the PLOSP, the lifespan of human beings can be fundamentally elongated. The PLOSP has been supported by a certain number of inadvertent facts in the past. Further verification of the PLOSP can be achieved with relatively simple studies. Currently, the improving living and nutritional conditions and improving medical standards can prolong human life. However, these measures require a relatively long process, and the extension of lifespan by these approaches also has limitations.

## Figures and Tables

**Figure 1 biology-11-00656-f001:**
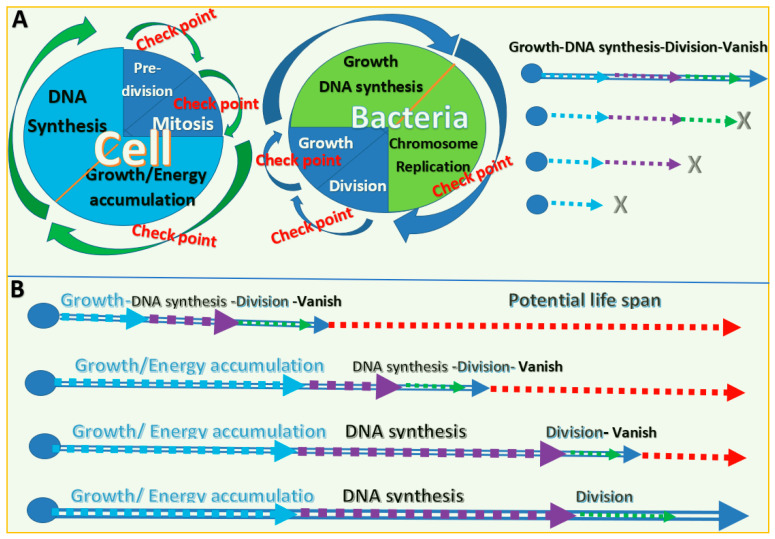
Life cycle and lifespan of a single cell. (**A**) The stages of the life cycle and check points for stopping the life cycles of a cell and bacteria. X indicates the abolishment or stopping of a development stage of a life cycle. (**B**) Possible methods to increase the lifespan of single cells. Different life stages are represented by arrows of different colors. Elongated arrow bars indicate the possible extension of a life stage. Red arrows indicate the potential maximum lifespan of the single cell.

**Figure 2 biology-11-00656-f002:**
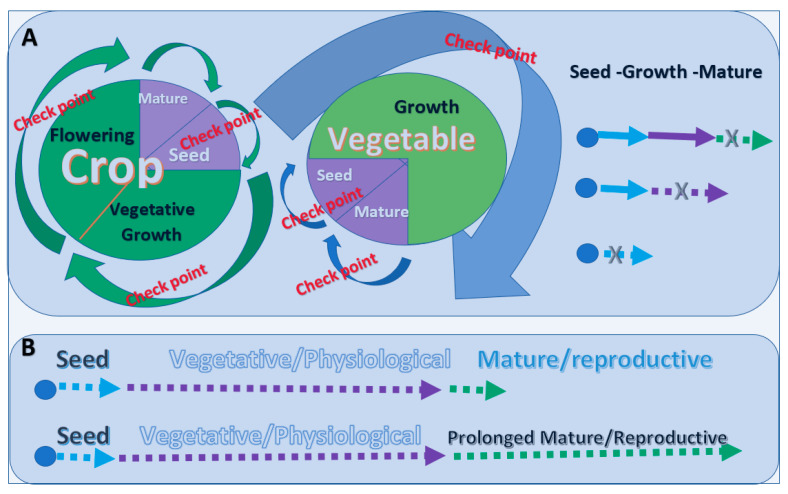
Life stages and lifespan of plants. (**A**) Stages of the life cycles of crops and vegetables. Stopping any stage during the growth of a plant will lead to its death. (**B**) Meaningful elongation of the stages of the life cycle to extend the lifespan of plants. A plant can increase its lifespan by increasing the period of vegetative growth. A vegetable can increase its lifespan by having a long period of reproduction.

**Figure 3 biology-11-00656-f003:**
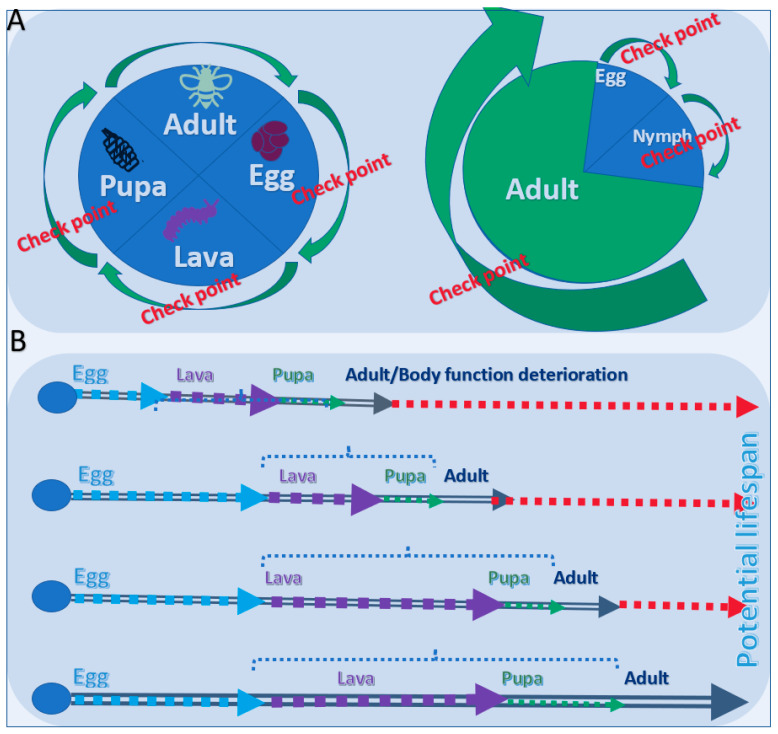
Life stages and lifespan of insects. (**A**) Four stages of the life cycle of most insects (left) and three stages of the life cycle represented by a grasshopper (right). Stopping any of these growth stages will kill the insect. (**B**) Extension of dormancy of egg, slow-down the growth of lava (or nymph), and maturation of pupa will elongate the lifespan.

**Figure 4 biology-11-00656-f004:**
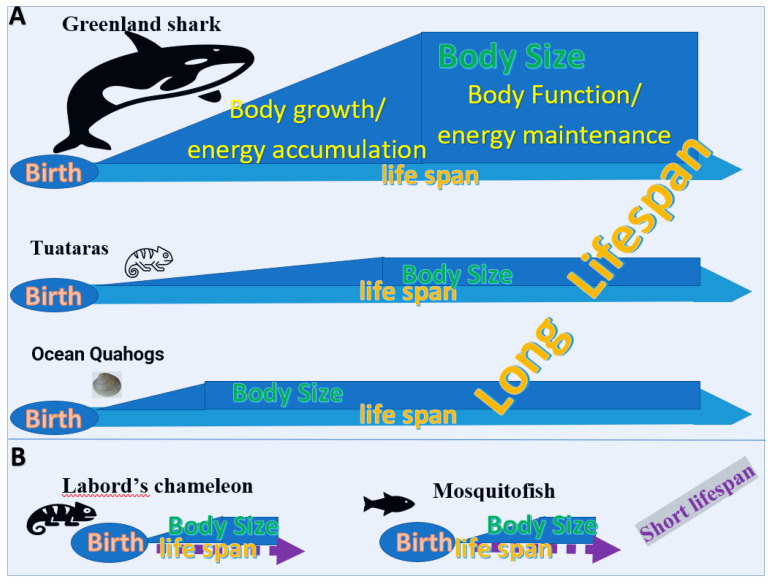
Examples of different lifespans in animals. (**A**) Three ways of achieving a long lifespan in the life cycle. The Greenland shark achieves a long lifespan by continued growth into a large body size, which therefore leads to a long reproductive period. Tuataras experience a long lifespan mainly by their slow body growth and long reproductive period. Ocean quahogs experience a long lifespan mainly due to the longevity of their reproductive activity. (**B**) Two examples of animals with short lifespans. Both Labord’s chameleons and mosquitofish have short lifespans because of their rapid body maturation and short reproductive period.

**Figure 5 biology-11-00656-f005:**
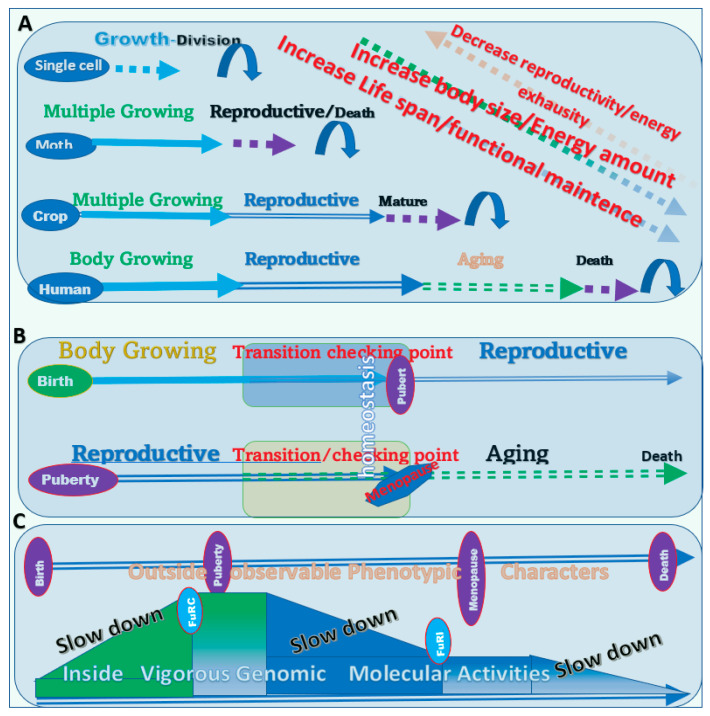
Rules of longevity and essential checkpoints for a long lifespan. (**A**) In the kingdoms of life on earth, lifespan is highly positively associated with body size and negatively associated with productivity. (**B**) The unrecognized transition period between key life stages. There should be transition periods between body growth and body maturity/reproductivity and between reproductivity and post-productivity or the aging stage. (**C**) The changes in life stages are the result of accumulative genomic activities that are not observable by the human eye.

**Figure 6 biology-11-00656-f006:**
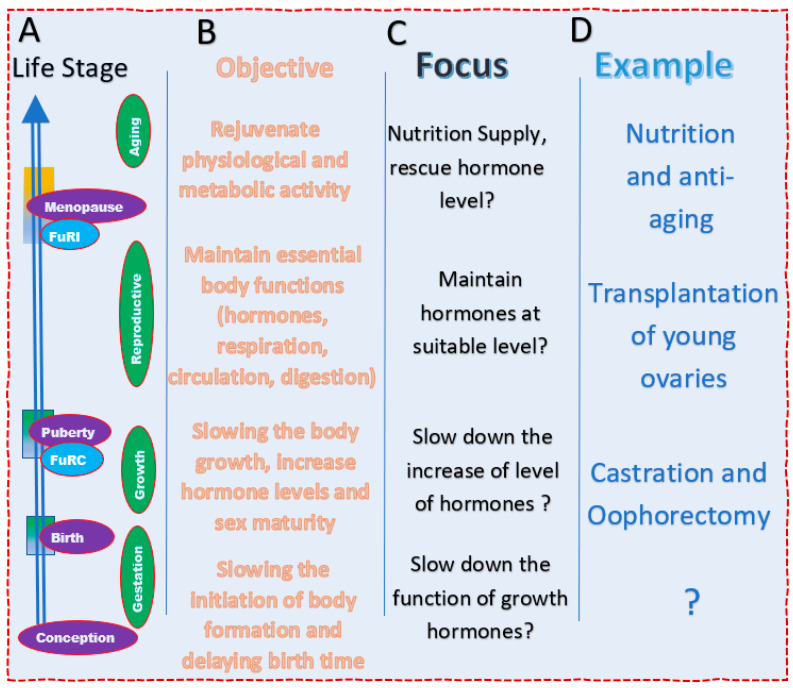
Strategies of lifespan extension based on the principal law of lifespan. (**A**) The key transitional points of the life cycle. (**B**) The objectives of different life stages. (**C**) Strategic focus points in different life stages. (**D**) The example that may be pursued to support the strategies.

## Data Availability

Not applicable.

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
