# Peer review of "Healthy Long-Lived Human Beings—Working on Life Stages to Break the Limitation of Human Lifespans"

_biology, 2022, doi:10.3390/biology11050656_

Round 1

Reviewer 1 Report

The topic of the manuscript is of very broad scientific interest. The illustration of the author's hypothesis via figures facilitates the perception of the findings by readers. The presented hypothesis is novel and presented in a clear and detailed manner. However, in my opinion, several issues need to be addressed:

  • Since the manuscript presents a novel hypothesis that is to a large extent not yet supported by experimental results, it is appropriately submitted as a perspective/hypothesis. However, some statements regarding expected results are rather strongly expressed. I suggest modifying them to emphasise that this is a hypothesis. For example: "The ones living longer group should be the one before puberty. "
  • Overall, many statements referring to previous research are not supported by citations. Please ensure that all statements referring to previous research are supported by citations. For example, lines 46-47, lines 88-91, and lines 320-352. However, these are just examples.
  • The text requires additional editing due to some grammatical omissions. For example, "lessens" has been used instead of "lessons".
  • You refer to PLOSP as a known fact without specifying whether it is a novel hypothesis developed by you and presented for the first time here, or whether you or other authors have already introduced this hypothesis before. Please provide more details when introducing the term for the first time in the introduction.

Author Response

Reviewer #1.

I appreciate profoundly for your advocate for such an important issue. With your support, we may make the history on the biomedical research. I feel that eventually the biomedical community will recognize the PLOSP and a new direction in the filed will be open.  I will answer your comments point by point.

The topic of the manuscript is of very broad scientific interest. The illustration of the author's hypothesis via figures facilitates the perception of the findings by readers. The presented hypothesis is novel and presented in a clear and detailed manner. However, in my opinion, several issues need to be addressed:

  • Since the manuscript presents a novel hypothesis that is to a large extent not yet supported by experimental results, it is appropriately submitted as a perspective/hypothesis. However, some statements regarding expected results are rather strongly expressed. I suggest modifying them to emphasise that this is a hypothesis. For example: "The ones living longer group should be the one before puberty. "

A1.  Thank you so much for thoughtful suggestion.   I may have been too excited and feel too confident. I have made changes in several places to low down the tone by adding phrases such as “most likely”, “probably”, “may”, “possibly”, “It is expected” “very likely”, etc..  Please see the copy of manuscript “with tracked changes” for all the changes.

  • Overall, many statements referring to previous research are not supported by citations. Please ensure that all statements referring to previous research are supported by citations. For example, lines 46-47, lines 88-91, and lines 320-352. However, these are just examples.

A2. I understand your concern.  There are many publications on the life stages, although I do not feel that they are really much related to this publication. Nevertheless, I have added references in these three places as well as about ten other places. Together 15 more reference have been added throughout the manuscript.

  • The text requires additional editing due to some grammatical omissions. For example, "lessens" has been used instead of "lessons".

A3:   As you see, errors have been corrected in multiple places.

  • You refer to PLOSP as a known fact without specifying whether it is a novel hypothesis developed by you and presented for the first time here, or whether you or other authors have already introduced this hypothesis before. Please provide more details when introducing the term for the first time in the introduction.

A4. Thanks for reminding me on this point. This is the first time for me to introduce this hypothesis. I have modified the sentences in the introduction to reflect this point.

Reviewer 2 Report

The Author described the limits of current biomedical approach in searching extension of lifespan and recognized the need of a new direction based on the principal law of lifespan (PLOSP) by the proper manipulation of well-defined growing stages of the lifespan, indicating that modulating the biological functions at the early stage of life may open a new path to extend lifespan. The review is complete, evidences are solid, and ideas are novel, logic, and refresh.

The manuscript needs very minor revision before it is published:

Line 103, The “e” of environment should be capitalized “E”.

Line 120, delete “of” after “stage”

Line 122, “by increase the period of vegetative growth” should be “by increasing the period of vegetative growth”.

Line 179, “Three ways of being a lone” should be “Three ways of being a long”

Line 227, “Saving am aged person’s life is similar…” should be “Saving an aged person’s life is similar”

Line 398, “our current human lifespan”, it is probably not a bad idea to add “average” in front of “human” because it is in fact the average lifespan.

Author Response

A:  If this publication makes a history in the biomedical research, you will feel proud of being a reviewer for it and a person among people supporting the PLOSP.

Please see attached the copy with tracked changes and a clean copy. I appreciate so much for your careful reading the manuscript.  These errors have been corrected.

The Author described the limits of current biomedical approach in searching extension of lifespan and recognized the need of a new direction based on the principal law of lifespan (PLOSP) by the proper manipulation of well-defined growing stages of the lifespan, indicating that modulating the biological functions at the early stage of life may open a new path to extend lifespan. The review is complete, evidences are solid, and ideas are novel, logic, and refresh.

The manuscript needs very minor revision before it is published:

Line 103, The “e” of environment should be capitalized “E”.

Line 120, delete “of” after “stage”

Line 122, “by increase the period of vegetative growth” should be “by increasing the period of vegetative growth”.

Line 179, “Three ways of being a lone” should be “Three ways of being a long”

Line 227, “Saving am aged person’s life is similar…” should be “Saving an aged person’s life is similar”

Line 398, “our current human lifespan”, it is probably not a bad idea to add “average” in front of “human” because it is in fact the average lifespan.